# An Intrabody against B-Cell Receptor-Associated Protein 31 (BAP31) Suppresses the Glycosylation of the Epithelial Cell-Adhesion Molecule (EpCAM) via Affecting the Formation of the Sec61-Translocon-Associated Protein (TRAP) Complex

**DOI:** 10.3390/ijms241914787

**Published:** 2023-09-30

**Authors:** Tianyi Wang, Changli Wang, Jiyu Wang, Bing Wang

**Affiliations:** College of Life Science and Health, Northeastern University, 195 Chuangxin Road, Hunnan District, Shenyang 110819, China; wangtianyi@mail.neu.edu.cn (T.W.); 1710070@stu.neu.edu.cn (C.W.); 2101371@stu.neu.edu.cn (J.W.)

**Keywords:** BAP31, EpCAM, intrabody, glycosylation, gastric cancer

## Abstract

The epithelial cell-adhesion molecule (EpCAM) is hyperglycosylated in carcinoma tissue and the oncogenic function of EpCAM primarily depends on the degree of glycosylation. Inhibiting EpCAM glycosylation is expected to have an inhibitory effect on cancer. We analyzed the relationship of BAP31 with 84 kinds of tumor-associated antigens and found that BAP31 is positively correlated with the protein level of EpCAM. Triple mutations of EpCAM N76/111/198A, which are no longer modified by glycosylation, were constructed to determine whether BAP31 has an effect on the glycosylation of EpCAM. Plasmids containing different C-termini of BAP31 were constructed to identify the regions of BAP31 that affects EpCAM glycosylation. Antibodies against BAP31 (165–205) were screened from a human phage single-domain antibody library and the effect of the antibody (VH-F12) on EpCAM glycosylation and anticancer was investigated. BAP31 increases protein levels of EpCAM by promoting its glycosylation. The amino acid region from 165 to 205 in BAP31 plays an important role in regulating the glycosylation of EpCAM. The antibody VH-F12 significantly inhibited glycosylation of EpCAM which, subsequently, reduced the adhesion of gastric cancer cells, inducing cytotoxic autophagy, inhibiting the AKT-PI3K-mTOR signaling pathway, and, finally, resulting in proliferation inhibition both in vitro and in vivo. Finally, we clarified that BAP31 plays a key role in promoting N-glycosylation of EpCAM by affecting the Sec61 translocation channels. Altogether, these data implied that BAP31 regulates the N-glycosylation of EpCAM and may represent a potential therapeutic target for cancer therapy.

## 1. Introduction

Gastric cancer (GC) is one of the most common carcinomas and is the fourth leading cause of cancer-related deaths globally [1,2]. Prognoses are poor, and recurrence rates are high for the patients with GC due to tumour progression [3]. Thus, it is vitally important to elucidate the underlying molecular mechanisms and develop effective targeted therapies for GC treatment.

The GC patients who overexpressed the epithelial cell-adhesion molecule (EpCAM) exhibited a lower five-year overall survival rate than the EpCAM-negative patients [4,5,6,7]. Tumor-specific mutations have not been reported in EpCAM so far, while hyperglycosylation of EpCAM at N74, N111, and N198 in carcinoma was commonly seen compared to normal epithelia [8,9,10]. With carcinoma-derived EpCAM being heavily glycosylated, this may result in a molecule with improved stability, cell surface presence, and enhanced signaling capacity [10]. 

The nascent polypeptide is transferred to the Sec61 translocation channel on the endoplasmic reticulum (ER) membrane and, then, cotranslationally translocated into the ER lumen [11]. During this process, if the polypeptide contains the sequence Asn-Xaa-Ser/Thr (where Xaa denotes any amino acid except Pro), it may be N-glycosylated by the multimembrane protein complex oligosaccharyltransferase (OST) [12]. Therefore, it is a key intermediate step in the glycoprotein modification process that new peptides enter the Sec61 pore structure of the ER membrane and transfer to the ER lumen. Our previous study demonstrated that BAP31 colocalizes with Sec61β and translocating-chain-associated membrane protein (TRAM), which are integral membrane components of the ER Sec61 translocation channel, and regulates retrotranslocation of CFTRDF508 from the ER [13]. So far, it is unclear whether BAP31 is involved in the formation of the Sec61 translocation channel, affecting the glycosylation of EpCAM, and resulting in an anticancer role.

BAP31, a resident integral protein of the ER membrane, associates with newly synthesized integral membrane proteins and regulates the exportation of selected membrane proteins from the ER to downstream compartments of the secretory pathway [13]. BAP31 also functions in quality control and sorting of a number of client membrane proteins [14,15,16,17,18]. In addition, our recent studies clarified that BAP31 is involved in activation of T cells [19], the formation of amyloid-β proteins [20], hepatic lipid accumulation [21], and regulating the expression of valosin-containing protein [22]. Recent studies have indicated that BAP31 is a novel cell surface marker for human embryonal carcinoma cells [23,24,25]. Increasing evidence has showed that expression levels of BAP31 are dramatically increased in different solid malignancies, such as cervical cancers [26], melanoma [27], and colorectal cancers [28]. We previously found that BAP31 is highly expressed in gastric intestinal type adenocarcinoma compared to normal gastric mucosa [29]. Furthermore, we analyzed the relationship of BAP31 with 84 kinds of tumor-associated antigens and found that overexpression of BAP31 in gastric cancer cell line MKN-45 increased the protein levels of EpCAM [29]. 

Therefore, for the first time, we explored the impact of BAP31 on glycosylation modifications of EpCAM. Additionally, a single-domain antibody against BAP31 was screened from a human phage antibody library and found to interfere with the formation of Sec61 and auxiliary translocon-associated protein (TRAP) complex, leading to significant anti-tumor effect both in vitro and in vivo by inhibiting N-glycosylation of EpCAM. 

## 2. Results

### 2.1. BAP31 Has an Effect on the Protein Level of EpCAM

We previously analyzed the relationship of BAP31 with 84 kinds of tumor-associated antigens and found that when BAP31 was overexpressed in the gastric cancer cell line MKN-45, protein levels of EpCAM were increased significantly [29]. Therefore, we want to investigate whether there is an interaction between BAP31 and EpCAM. Endogenous BAP31 and EpCAM were coimmunoprecipitated from the lysate of the gastric cancer cell line MKN-45 in the hydrophobic surfactant Nonidet P 40 (NP40) (Figure 1A). The colocalization of EpCAM with BAP31 in the ER was observed by confocal microscope. Intracellular staining shown that EpCAM had significant ER localization and was overlaid with BAP31 in the ER (Figure 1B).

The effect of BAP31 on expression levels of EpCAM in GC cell lines MKN-45 and AGS was investigated. First, the mRNA level of EpCAM remained nearly constant after BAP31 overexpression or knockdown in both cells (data not shown). Then, we investigated protein levels of EpCAM by Western blotting. The nonglycosylated band of EpCAM is generally at 35 kDa but is rarely detected in cells. The commonly detected bands in cells with a molecular weight (MW) of 40–42 kDa are the glycosylated EpCAM (Appendix A). When BAP31 was knocked down by siRNA, the glycosylated EpCAM were significantly decreased in both MKN-45 and AGS cells. When BAP31 was overexpressed by transfecting the pcDNA3.1(-)-BAP31-Flag plasmid into the two cell lines, the glycosylated EpCAM were increased significantly (Figure 1C,D), indicating that BAP31 is positively correlated with the protein levels of EpCAM in GCs. Flow cytometry was used to investigate the effect of BAP31 on the cell surface EpCAM of MKN-45 cells. Both cell surface and total EpCAM levels were decreased when BAP31 was knocked down (Figure 1E,F), and cell surface EpCAM was decreased more significantly compared to total EpCAM (68.5%, *p* < 0.001 vs. 43.2%, *p* < 0.01), indicating that BAP31 has a major effect on the cell surface EpCAM. Because hyperglycosylation efficiently increases EpCAM’s stability and cell surface retention [9,10,11], we hypothesized that BAP31 may have an effect on promoting the glycosylation of EpCAM.

### 2.2. The Effect of BAP31 on N-Glycosylated EpCAM

Asparagine at amino acids N111, N76, and N198 is reported to be glycosylated, and triple-mutant EpCAM N76/111/198A is no longer modified by glycosylation [10]. Point mutations were sequentially inserted into the domain of wild type (WT) EpCAM to exchange the three asparagine glycosylation sites for alanines to generate the triple mutant (N74/111/198A). The effect of BAP31 on protein levels of WT-EpCAM and mutant EpCAM were examined by Western blotting. Treatment of HEK-293 cells with glycosidase (PFNGase-F) made the WT-EpCAM with an MW of 40 kDa migrate equal to nonglycosylated EpCAMN76/N111/N198A with an MW of 35 kDa (Figure 2A, lane 5 and 6, Figure 2B). When BAP31 was overexpressed, protein levels of N-glycosylated EpCAM were significantly increased (Figure 2A, lane 1 and 2); however, nonglycosylated EpCAMN76/N111/N198A was unaffected (Figure 2A, lane 3 and 4, Figure 2B). When BAP31 was knocked down by siRNA, protein levels of WT-EpCAM were significantly decreased; however, nonglycosylated EpCAMN76/N111/N198A was unaffected (Figure 2C,D), indicating that BAP31 only increases protein levels of glycosylated EpCAM but has almost no effect on nonglycosylated EpCAM.

Because BAP31 has a hydrophobic N-terminal transmembrane region and a cytoplasmic C terminal region (Appendix A), the C terminus of BAP31 was segmentally excised, as illustrated in Figure 2E. Then, plasmids containing a different C-terminus were transfected into MKN-45 cells, and protein levels of N-glycosylated EpCAM were investigated. Expression of full-length BAP31 significantly increased N-glycosylated EpCAM. BAP31 (1–205 aa) exhibited the same promotion effect as full-length BAP31; however, BAP31 (1–164 aa) had little effect on N-glycosylated EpCAM (Figure 2F,G), indicating that the amino acid region from 165 to 205 in BAP31 plays an important role in regulating the N-glycosylated EpCAM.

### 2.3. Screening an Anti-BAP31 Human Single-Domain Antibody and Identifying Its Effect on Glycosylated EpCAM

To further investigate whether the 165–205 aa of BAP31 is essential for the N-glycosylation of EpCAM, we screened antibodies against BAP31 (165–205 aa) from a human phage single-domain antibody library. VH-F12 were obtained that strongly bound to BAP31 through ELISA (Appendix A). The coding sequences of VH-F12 were cloned into pcDNA3.1(-) or the lentivirus vector (pLVX-IRES-ZsGreen1) with an HA-tag (YPYDVPDYA) and an ER retention signal (KDEL) in the C terminal region, as well as an ER-guided peptide in the N terminal. 

Expression levels of VH-F12 were evaluated by immunofluorescence and flow cytometry using an anti-HA monoclonal antibody (HA-7). MKN-45 cells were transduced with 50 MOI lentivirus and delivered a BAP31-specific antibody or, as a control with vector alone (Lenti-ZsGreen). Lenti-F12 showed high transduction efficiency of greater than 75% in MKN-45 cells 72 h post-transduction (Appendix A). The transfection efficiency of pcDNA3.1(-) -VH-F12 was examined by flow cytometry analysis. Expression of VH-F12 reached approximately 70% in MKN-45 cells following 96 h of transfection (Appendix A). 

### 2.4. VH-F12 Colocalizes with BAP31 in the ER and Suppresses N-Glycosylated EpCAM

The association between BAP31 and the VH-F12 intrabody was evaluated in HEK-293 cells by IP analysis. The pcDNA3.1(-)-VH-F12, pcDNA3.1(-)-VH-irrelevant (S5), pcDNA3.1(-)-BAP31-Flag and vector-alone plasmids were transiently transfected in HEK-293 cells individually or together. In addition, to evaluate whether VH-F12 combines with BAP31 in the specific C terminal region (165–205 aa), we constructed a BAP31 (1–164 aa)-Flag and BAP31 (1–205 aa)-Flag plasmids that were transfected with HEK-293 cells individually or together as indicated. VH-F12 specifically bound to BAP31 (1–205 aa), but not to BAP31 (1–164 aa), confirming that VH-F12 combines to the (165–205 aa) region of BAP31 (Figure 3A). 

To investigate whether the intrabody localizes to the ER and combines with BAP31 in the same compartment, we transfected MKN-45 cells with pcDNA3.1 (-)-VH-F12-HA or control pcDNA3.1(-)-VH-irrelevant (S5)-HA. VH-irrelevant (S5) was constructed with a sequence encoding an HA-tag and a nuclear localization sequence at the C terminus (as described in Methods). As expected, the VH-F12 intrabody showed significant intracellular accumulation and colocalized with BAP31 in the ER, while the intrabody control (S5) did not combine with BAP31 (Figure 3B).

The effect of VH-F12 on expression levels of glycosylated EpCAM was evaluated. VH-F12 had a significant inhibitory effect on N-glycosylated EpCAM. In addition, protein levels of BAP31 and the other membrane protein integrin β1 were not significantly altered following VH-F12 expression (Figure 3C,D). Because EpCAM is a cell-adhesion molecule, the adhesion of MKN-45 to the extracellular matrix was investigated using Matrigel-coated plates, and the number of adherent MKN-45 cells was quantified by the number of crystal violet-stained cells. The adhesion of MKN-45 to the extracellular matrix was significantly decreased following VH-F12 expression (Figure 3E,F). EpCAM on the cell surface of VH-F12 expressing cells was decreased by approximately 65.8% (*p* < 0.001), and total EpCAM was decreased by approximately 41.3% by flow cytometry (*p* < 0.01), indicating that VH-F12 has an effect primarily on N-glycosylated EpCAM (Figure 3G,H). In addition, we observed the effect of VH-F12 on EpCAM in BAP31 stable knockout MKN-45 cells. When BAP31 was knocked out by sh-RNA, N-glycosylated EpCAM was significantly decreased, but the extra expression of VH-F12 did not enhance this reduction (Appendix A), indicating that VH-F12 participates in the inhibition of EpCAM through BAP31.

### 2.5. VH-F12 Intrabody Promotes MKN-45 Cell Death In Vitro and In Vivo 

We inferred that VH-F12 might have an anti-tumour effect by inhibiting the glycosylation of EpCAM. General cytotoxicity from lentivirus transduction and cytotoxicity related to VH expression were evaluated. The effect of Lenti-VH-F12 on the viability of MKN-45 cells was evaluated at different times post-transduction at 50 MOI. As a control, MNK-45 cells were transduced with Lenti-irrelevant (S5) or with a Lenti-control alone. VH-F12 intrabody induced approximately 54% inhibition of proliferation following 96 h of transduction. In contrast, Lenti-VH-S5 or the Lenti-control group did not exhibit any proliferation inhibition effects on MKN-45 cells (<10%) (Appendix A). Because low cytotoxicity of intrabodies is usually observed at 50 MOI, we selected this concentration for further analysis in vivo. 

Next, we sought to investigate whether the decreased cell viability is associated with induced apoptosis through flow cytometry by Annexin V/PI staining. MKN-45 cells were transfected with pcDNA3.1(-)-VH-F12, pcDNA3.1(-)-irrelevant (S5) or vector alone. At 72 h post-transfection, VH-F12 led to a significantly increased total percentage of dead cells compared to the control group. However, no significant difference was observed in the percentage of early apoptotic cells (Annexin positive but PI negative) (Figure 4A,B).

We evaluated the levels of apoptosis-related proteins, Bax, Bak, Bad, Bid, caspase 3, caspase 9, and PARP. However, no significant differences in these proteins were observed in VH-F12 expressing MKN-45 cells compared to control cells (data not shown). Taken together, these data indicate that the increased death rate is not primarily due to the induction of apoptosis, but, rather, non-apoptotic death in GC cells. 

Punctate staining of LC3, a marker of autophagosomes, was observed in VH-F12 expressing MKN-45 cells but not in VH-S5 or vector-transfected cells (Figure 4C), suggesting that VH-F12 activated autophagy. The activation of the autophagy was further confirmed by Western blotting for autophagy-related proteins. Protein levels of Beclin1, ATG5, LC3α, and LC3β were significantly increased in VH-F12 expressing MKN-45 cells compared to levels in control cells (Figure 4D,E), indicating that VH-12 intrabody has a role in autophagy activation.

We measured viability of VH-F12 expressing MKN-45 cells in response to 10 mM autophagy inhibitor (3-Methyladenine, 3-MA) treatment for 24 h. The viability of VH-F12 expressing cells was remarkably recovered compared to cells without 3-MA treatment (Figure 4A), indicating that VH-F12 may play an anti-oncogenic role by inducing autophagic cell death. Because deglycosylated EpCAM inhibited proliferation through activating autophagy by suppressing Akt/mTOR signaling pathway [30], we investigated whether this pathway was inhibited when BAP31 was suppressed. The PI3K/Akt/mTOR pathway was significantly inhibited when BAP31 was knocked down, and VH-F12 induced a similar effect as BAP31 knockdown with respect to this pathway (Appendix A).

To evaluate the effect of the VH-F12 intrabody on tumor growth in vivo, we used an MKN-45 cell xenograft mouse model. Two group of mice were inoculated with MKN-45 cells. When tumors reached a volume of approximately 150 mm^3^, one group of mice was given multipoint i.t. injection of lentivirus expressing VH-F12. After two lentivirus-VH-F12 treatments, only small tumors were macroscopically visible near the right leg. In contrast, much larger tumors were visible in mice in the negative control group (Figure 4F). Tumor volume and weight of the VH-F12-expressing group was significantly lower compared to the control group (Figure 4G,H). 

### 2.6. BAP31 Is Essential for the Formation of Sec61 and TRAP Complex

N-glycosylation usually begins with the entry of translocators into the ER cavity during the translation of new peptide chains. In mammalian cells, one-third of all polypeptides are transported into or across the ER membrane via the Sec61 channel, while the Sec61-TRAP complex supports translocation of a subset of precursors [31]. BAP31, an organizational pattern of ribosome/translocon complex, binds to Sec61β, TRAPα, and participates in the translocation process of newly synthesized proteins [32]. However, the detailed function of BAP31 was not demonstrated. The effect of BAP31 and VH-F12 on formation of Sec61 and Sec61-TRAP complexes was first analyzed using a native gel electrophoresis procedure. The BN-PAGE gels were probed with antibodies to Sec61α to determine Sec61 and Sec61-TRAP complexes, respectively, according to the reports [11,31,33]. The cleared cell lysate was resolved by BN-PAGE for protein immunoblot analysis. Surprisingly, Sec61 and Sec61-TRAP complexes were hardly detected in BAP31 knockout MKN-45 cells (Figure 5A,B), indicating that BAP31 is needed for the formation of both Sec61 and Sec61-TRAP complexes. However, VH-F12 reduced the formation of Sec61-TRAP, but not affected the formation of Sec61 complexes (Figure 5C,D).

## 3. Discussion

Approximately 30% of the cellular proteome enters the ER through the Sec61 protein-conducting channel and its associated proteins, which are collectively referred to as translocon. The translocon accepts nascent secretory and membrane proteins from the signal recognition particle and, together with the translating ribosome, cotranslationally directs their topology by partitioning the elongating polypeptide into the ER lumen, cytosol, and lipid bilayer [31]. ER is the most common N-glycosylated modification site in eukaryotic cells and BAP31 is a ubiquitously expressed ER membrane protein. We demonstrated for the first time that BAP31 has function in promoting N-glycosylation of EpCAM. Our data showed that BAP31 interacts with EpCAM in the ER and is positively correlated with N-glycosylated EpCAM in both gastric cancer cell lines and HEK-293 cells (Figure 1 and Figure 2). 

Because N-glycosylated EpCAM has good stability and extended half-life [10], we inferred that BAP31 may increase the protein level of N-glycosylated EpCAM by promoting its glycosylation modification. EpCAM is a glycoprotein which has three glycosylation sites. We generated triple mutants by exchanging three asparagine glycosylation sites for alanines. EpCAM N76/111/198A was no longer modified by the addition of glycosyl residues. When BAP31 was overexpressed in HEK-293 cells, protein levels of glycosylated EpCAM increased correspondingly, however, protein levels of EpCAMN76/111/198A were negligibly affected (Figure 2A,C). These data showed that BAP31 has no promoting function for EpCAM that cannot be glycosylated, suggesting that BAP31 may play a role in protein glycosylation. 

Translocon accessory factors adjacent to the Sec61 channel are responsible for covalent attachment of N-linked carbohydrates to some protein by OST [34,35]. Translocation is facilitated by the TRAP [36,37]. Their roles are not understood in detail. Recent studies suggest that TRAP may adopt a chaperone-like function in helping to assemble the correct topology for polytopic membrane proteins [36]. While the Sec61 complex facilitates translocation of all polypeptides with amino-terminal signal peptides or transmembrane helices, the Sec61-TRAP complex supports translocation of only a subset of precursors [31]. Previous study has found that BAP31 is an organizational pattern of ribosome/translocon complex [32], however, the detailed function of BAP31 was not demonstrated. Our results showed that BAP31 affected the formation of Secc61 complex. When BAP31 was knocked out, Sec61 complex could hardly be formed, nor could Sec61-TRAP complex, indicating BAP31 plays a key role in maintaining the stability of the Sec61 complex. Initially, the VH-F12 plasmid was constructed to express and locate in the cytoplasm. However, the expression level of VH-F12 was very low in this situation, to the extent that bands were almost undetectable by Western blotting. We speculated that VH-F12 may be rapidly degraded in the cytoplasm, resulting in insufficient time to effectively bind to BAP31. Therefore, the signal peptide was added to guide VH-F12 into the ER lumen. Due to the translation, folding, and maturation of BAP31 occurring within the ER lumen, guiding VH-F12 into the ER lumen may greatly increase its chances of binding with BAP31 (165–205). The results confirmed that the addition of signal peptides enabled stable expression of VH-F12 in cells and promoted effective binding of VH-F12 to BAP31 (165–205) (Figure 3A). We also found that intrabodies targeting other C-terminal regions of BAP31 (124–164 aa or 206–246 aa) had little effect on the glycosylation of EpCAM (data not shown). Therefore, we speculate that the binding of VH-F12 to BAP31 (165–205) may affect the function of BAP31 by inhibiting its correct folding in the ER lumen. The incorrect three-dimensional conformation of BAP31 further affects the interaction between Sec61 and TRAP complexes. However, more research is needed on the specific mechanism by which VH-F12 functions. For example, constructing a BAP31 mutant plasmid that cannot effectively bind to VH-F12 and studying the effect of VH-F12 on EpCAM glycosylation by expressing this mutation BAP31 in BAP31 knockout cells can further validate our hypothesis.

Furthermore, our data demonstrate that EpCAM is the sorting substrate of Sec61-TRAP complexes. BAP31 probably affects the translocation channel of EpCAM into ER by regulating the formation of Sec61 and Sec61-TRAP complexes, and, then, affects the glycosylation of EpCAM. The possible mechanism of BAP31 participating in the process of EpCAM glycosylation is shown in Figure 6.

VH-F12 significantly decreased N-glycosylated EpCAM (Figure 3C,D), which led to reduced cell adhesion and cell surface EpCAM (Figure 3E–H). We next evaluated the anti-tumor effect of VH-F12 because cell surface EpCAM plays a major role in the recognition of receptors and ligands. We found that when VH-F12 was delivered by a lentivirus system, tumor cells exhibited reduced proliferation both in vitro and in vivo (Appendix A and Figure 4). In particular, VH-F12 induces tumor cell death by activating cytotoxic autophagy (Figure 4). Deglycosylated EpCAM enhances autophagy of cancer cells via PI3K/Akt/mTOR pathway in breast cancer cells [30]. The AKT-PI3K-mTOR pathway is activated in many types of cancers and inhibition of this signaling pathway is consistent with the reduced proliferation, colony formation, cell invasion ability, and increased cytotoxic autophagy [38,39,40]. Our data showed that BAP31 knockdown in gastric cancer cells was accompanied by the suppression of the PI3K/Akt/mTOR pathway, and VH-F12 induced a similar effect as BAP31 knockdown with respect to this pathway (Appendix A). Therefore, the results suggested that suppressing the N-glycosylation of EpCAM by VH-F12 induces autophagic cell death via PI3K/Akt/mTOR pathway in gastric cancer cells. Our study reveals a novel functional region of BAP31, and an intrabody against it inhibits N-glycosylation of EpCAM, activates genes related to cytotoxic autophagy, and inhibits the PI3K-Akt-mTOR pathway, significantly inhibiting tumor proliferation.

## 4. Material and Methods

### 4.1. Cell Lines and Animals

MKN-45, AGS and HEK-293 cell lines were obtained from the Cell Bank of the Chinese Academy of Sciences (Shanghai, China). The method of cell culture was performed according to previous our report [29].

### 4.2. Glycosidase Treatment

The cells were collected by adding trypsin and were subjected to centrifugation at 500× *g* for 10 min at 4 °C. The supernatant was removed, and pellets were resuspended in 0.5% SDS and 40 mM DTT. Following boiling for 10 min, samples were added into GlycoBuffer with 1% NP-40 and digested with PNGase (Calbiochem, La Jolla, CA, USA) at 37 °C for 4 h. Then, the samples were electrophoresed on a 5% polyacrylamide-SDS gel, transferred to a membrane, and reacted with an antibody against EpCAM.

### 4.3. Screening for Anti-BAP31 Human VH Single-Domain Antibody

The C-terminal peptide of BAP31 (GI: 550343), consisting of 42 amino acids, (165–205 aa) was synthesized (Wuhan Moon Biosciences, Wuhan, Hubei, China), and the purity of each polypeptide was greater than 98%. The method of screening was performed according to our previous report [29]. A peptide of MDM2 (GI: 260080636), consisting of 41 aa (234–274 aa), was synthesized by Wuhan Moon Biosciences. An irrelevant VH intrabody against MDM2 was used as the intrabody control, and VH-irrelevant (S5) was screened according to the method mentioned above.

### 4.4. Construction of the Plasmid

VH coding regions were excised by digestion with BamHI and XhoI and ligated into pcDNA3.1(-) or the lentiviral vector pLVX-IRES-ZsGreen1 vector. The VH coding regions were flanked by a sequence encoding an HA-tag (YPYDVPDYA) and an ER retention signal (KDEL) at the C terminus, as well as an ER guidance sequence in the N terminal region. VH coding regions were excised by digestion with BamHI and XhoI, and ligated into pcDNA3.1(-) or the lentiviral vector pLVX-IRES-ZsGreen1 vector. VH-irrelevant (S5) was constructed by a sequence encoding an HA-tag and nuclear localization sequence at the C terminus.

### 4.5. Lentiviral Transduction of the GC Cell Line

Lentivirus not expressing VH intrabody (Lenti-ZsGreen) for transduction control was also constructed. The lentivirus packaging systems pLVX-IRES-ZsGreen1, psPAX2, and pMD2.G were purchased from Addgene (Addgene, Cambridge, MA, USA). The method of lentiviral transduction was performed according to our previous report [29].

### 4.6. Plasmid or RNA Interference Transfection

The detailed plasmid transfection procedure was performed according to our previous report [29]. Three short interfering RNA (siRNA) duplexes were designed and purchased from GenePharma (Jiang Su, China). The cells were transfected with a mixture of three siRNA duplexes using Lipofectamine 2000 according to the instructions from GenePharma. After 72 h, the cells were harvested and used for Western blotting. A scrambled negative control 5′-TTCTCCGAACGTGTC-ACGUTT-3′ was used to mock transfect cells. *Homo*-BAP31-siRNA-1 targeted the sequence 5′-CCUCCAAUGAAGCCUUUAATT-3′, *Homo*-BAP31-siRNA-2 targeted the sequence 5′-GCGCAAAUUCGGAAGUAUTT-3′, and BAP31-siRNA-3 targeted the sequence 5′-GCGCGAAAUUCGGAAGUAUTT-3′. All sequences are located in the coding region.

### 4.7. Analysis of BAP31/Intrabody Colocalization by Immunofluorescence

HEK-293 cells were seeded onto glass cover slips and transfected with pcDNA 3.1(-)-VH-F12 or with the intrabody control, pcDNA 3.1(-)-VH-irrelevant (S5). The detailed method was performed according to our previous report [29]. The antibody sources and dilutions are listed in Appendix A.

### 4.8. Immunoprecipitation (IP) and Immunoblots 

Mouse gastric tissue extracts or MKN-45 cells were lysed in buffer containing 1% NP40 or 1% 3-(3-cholamidopropyl dimethylammonium)-1-propanesulfonate, 150 mM NaCl, 50 mM Tris-HCl (pH 7.4) and protease inhibitors. Lysates were assayed for protein content using the Bio-Rad reagent. For immunoprecipitation, after preclearing for 30 min with Protein-G Sepharose, lysates were incubated with antibodies for 90 min at 4 °C. Immune complexes bound to Protein-G Sepharose were recovered in a microfuge, washed four times with lysis buffer, and eluted in SDS sample buffer. The samples were analyzed by Western blotting. 

### 4.9. Apoptosis Analysis 

MKN-45 cells were transfected with pcDNA3.1(-)-VH-F12-HA or pcDNA3.1(-)-VH-irrelevant (S5)-HA, or were mock-transfected for 72 h. The apoptosis rate was determined by flow cytometry using an Annexin V/propidium iodide (PI) kit (Sigma-Aldrich, St. Louis, MO, USA). The percentages of cells stained with Annexin V, but not PI, were calculated as early apoptosis, while those of cells stained with both Annexin V and PI were calculated as the total percentages of dead cells. 

### 4.10. In Vivo Anti-Tumor Assay

All animal studies were performed according to guidelines approved by the Institutional Review Board of the College of Life and Health Sciences, Northeastern University, and conformed to guidelines for the ethical use of animals. Twelve BALB/c male nude mice (18–20 g each) were randomly divided into two groups and treated with subcutaneous 5 × 10^6^ MKN-45 cells. When the resulting tumors reached approximately 150 mm^3^, mice in the experimental group received intratumoral (i.t.) multipoint injection of 100 μL 1 × 10^7^ TU lentivirus-VH-F12 (*n* = 6). The i.t. injections were given twice three days apart. Mice in the negative control group received equivalent inactivated virus given in identical volume (*n* = 6). During the treatment, mice were weighed, and tumor volumes (V) were measured every three days using the following formula: V (mm^3^) = width (mm^2^) × length (mm) × 0.5. After three weeks, mice were euthanized, and tumor tissues were dissected from each mouse. 

### 4.11. Blue Native Gel Electrophoresis

The Sec61 and TRAP complexes were resolved by BN-PAGE. Briefly, the BN-PAGE resolving gel was cast as a 6–13% polyacrylamide gradient in 50 mM Bis-Tris-HCl (pH 7.0) and 500 mM aminocaproic acid and overlaid with a 4% stacking gel. Cells were lysed at 4℃ by a 30 min incubation with 20 mM Tris-Cl (pH 7.6), 150 mM NaCl, 5 mM Mg(OAc)_2_, 3 mM MnCl_2_, 2% digitonin, DNase I, and 1 × protease inhibitor cocktail. Cell lysates were clarified by centrifugation (10 min at 13,000× *g*). A supernatant fraction containing 200 µg of protein was mixed with 15% glycerol, 2 mM DTT, and 1/40 volume of 5% Coomassie Blue G250 in 500 mM 6-aminohexanoic acid and loaded on the gel. The gels were run at 4 °C for 16 h at 70 V with a cathode buffer of 0.02% Coomassie Blue G250 in 50 mM Tricine and 15 mM Bis-Tris, pH 7.0, and an anode buffer of 50 mM Bis-Tris, pH 7.0. After 16 h the cathode buffer was changed to contain 0.002% Coomassie Blue G250 and electrophoresis was continued at 500 V until the dye front reached the end of gel. A native protein gel marker (Life Technologies Corporation, Gaithersburg, MD, USA) was electrophoresed on gels to obtain apparent molecular weights of protein complexes. The proteins were transferred by wet blotting onto PVDF membranes. Membranes were destained with methanol and the subunits were detected by protein immunoblotting [33,41]. 

### 4.12. Animal Management 

All animals were managed according to the Care and Use of Medical Laboratory Animals guidelines (Ministry of Health, Beijing, China) and all experimental protocols were approved by the Laboratory Ethics Committee of China Medical University.

### 4.13. Statistical Analyses 

All experiments were repeated at least three times. The data are presented as the means ± standard error of the mean (S.E.M.). The student’s *t*-test was used to determine the significance of results. A *p*-value < 0.01 was considered statistically significant (** *p* < 0.01, *** *p* < 0.001). 

## 5. Conclusions

BAP31 is positively correlated with the protein levels of EpCAM and has a major effect on the cell surface EpCAM in GCs. BAP31 only increases protein levels of glycosylated EpCAM but has almost no effect on nonglycosylated EpCAM. The amino acid region from 165 to 205 in BAP31 plays an important role in regulating the *n*-glycosylated EpCAM. VH-F12 combined to the (165–205 aa) region of BAP31 significantly inhibited glycosylation of EpCAM which, subsequently, regulated the adhesion of gastric cancer cells, inducing cytotoxic autophagy, inhibiting the AKT-PI3K-mTOR signaling pathway, and, finally, resulting in proliferation inhibition both in vitro and in vivo. BAP31 plays a key role in promoting *n*-glycosylation of EpCAM by affecting the Sec61 translocation channels.

## Figures and Tables

**Figure 1 ijms-24-14787-f001:**
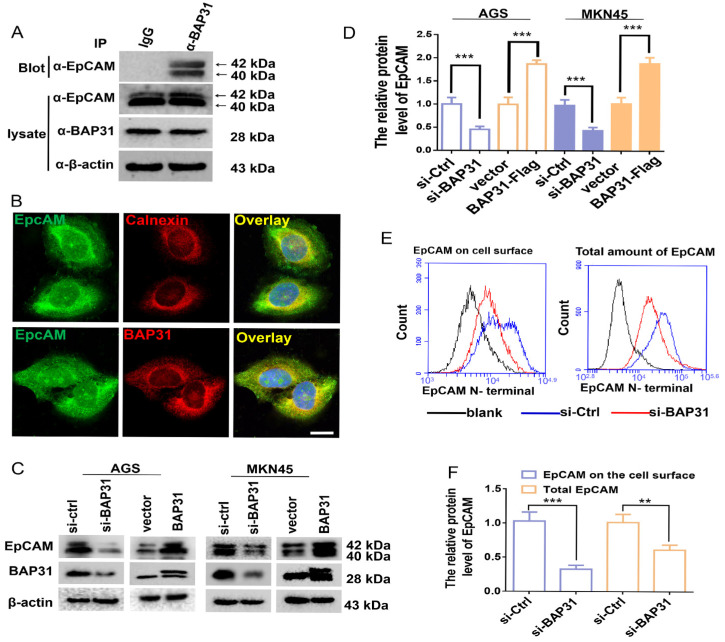
Association of BAP31 with EpCAM in the endoplasmic reticulum (ER). (**A**) Endogenous BAP31 and EpCAM were coimmunoprecipitated from MKN-45 lysates. Enriched membrane proteins from cell lysates (1% Nonidet P 40) were immunoprecipitated with an anti-BAP31 monoclonal antibody (7A3BB6) and examined with an antibody to EpCAM, which was compared to precipitates derived using preimmune IgG. (**B**) A confocal microscope was used to observe colocalization of BAP31 with EpCAM in the ER. EpCAM was visualized by incubation with the indicated primary antibody followed by an Alexa Fluor 488-conjugated (green) secondary antibody. The ER or BAP31 was visualized by incubating with a primary mouse antibody anti-calnexin (an ER marker) or anti-BAP31 followed by an Alexa 594-conjugated (red) secondary antibody. Yellow fluorescence indicates colocalization of EpCAM and BAP31 or calnexin immunoreactivity (scale bar, 5 μm). (**C**) Western blotting of expression levels of EpCAM in BAP31 knockdown or BAP31 overexpression AGS or MKN-45 cells. (**D**) Results are the means ± S.E.M. of three independent experiments, each performed in duplicate., *** *p* < 0.001; by student’s *t*-test. (**E**) Flow cytometry was used to assess the effect of BAP31 on EpCAM cell surface expression using an antibody against the N-terminus of EpCAM. (**F**) Results are the means ± S.E.M. of three independent experiments, each performed in duplicate. ** *p* < 0.01, *** *p* < 0.001; by student’s *t*-test.

**Figure 2 ijms-24-14787-f002:**
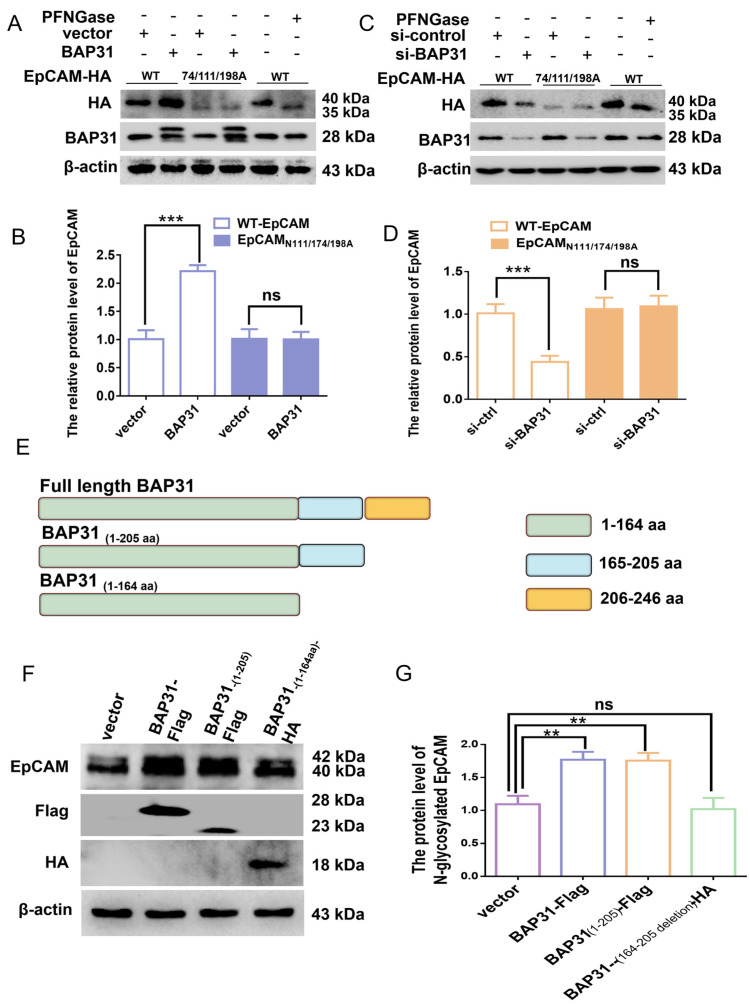
The effect of BAP31 on N-glycosylation of EpCAM. (**A**,**B**) Plasmids of EpCAMWT or EpCAMN74/111/198A or BAP31 were transfected into HEK-293 cells as indicated. The effect of BAP31 overexpression on protein levels of EpCAM was investigated. (**C**,**D**) Plasmids of EpCAMWT or EpCAMN74/111/198A or BAP31 were transfected into HEK-293 cells as indicated. The effect of BAP31 knockdown by siRNA on protein levels of EpCAM was investigated. (**E**) Three kinds of plasmids with different deletion of the C terminus of BAP31 were constructed as illustrated in the Figure. (**F**,**G**) The C-terminal half of BAP31 in the cytoplasm was segmented into three parts. The vector as indicated was transfected into MKN-45 cells, and protein levels of EpCAM were detected by Western blotting. Bar graphs were obtained by normalizing to actin. The results are the means ± S.E.M. of three independent experiments, each performed in duplicates. ** *p* < 0.01; *** *p* < 0.001, ns: no significance by student’s *t*-test.

**Figure 3 ijms-24-14787-f003:**
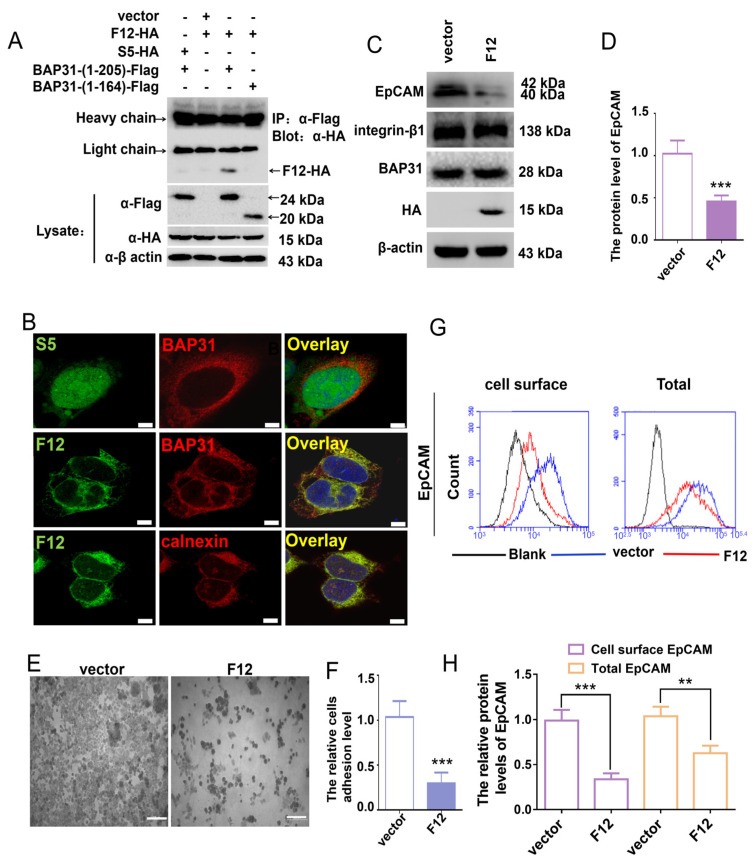
The effect of an antibody to BAP31 (VH-F12) on EpCAM. (**A**) HEK-293 cells were transiently transfected with plasmids as indicated. Immunoprecipitation was performed using an antibody against Flag, and immunoprecipitates were analyzed by Western blotting as indicated. (**B**) Cells were transfected with plasmids as indicated. VH-F12-HA and VH-S5-HA were visualized by incubation with the indicated primary monoclonal antibody (HA-7) to HA followed by an Alexa Fluor 488-conjugated (green) secondary antibody. BAP31 or ER was visualized by incubation with a primary polyclonal antibody to BAP31 or calnexin followed by an Alexa 594-conjugated (red) secondary antibody. Cell nuclei were stained with DAPI (blue) (scale bar, 5 μm). (**C**,**D**) MKN-45 cells were transfected with pcDNA3.1(-)-VH-F12-HA or a vector control. Following transfection for 72 h, Western blotting was used to analyze expression levels of EpCAM. (**E**,**F**) Control or VH-F12 expressing cells were incubated on a Matrigel-coated plate, and the number of adherent cells was visualized by crystal violet staining (scale bar, 200 μm). (**G**,**H**) The effect of VH-F12 on EpCAM on the cell surface was investigated by flow cytometry with an antibody against the N-terminus of EpCAM. Results are the means ± S.E.M. of three independent experiments, each performed in duplicate. ** *p* < 0.01, *** *p* < 0.001; by student’s *t*-test.

**Figure 4 ijms-24-14787-f004:**
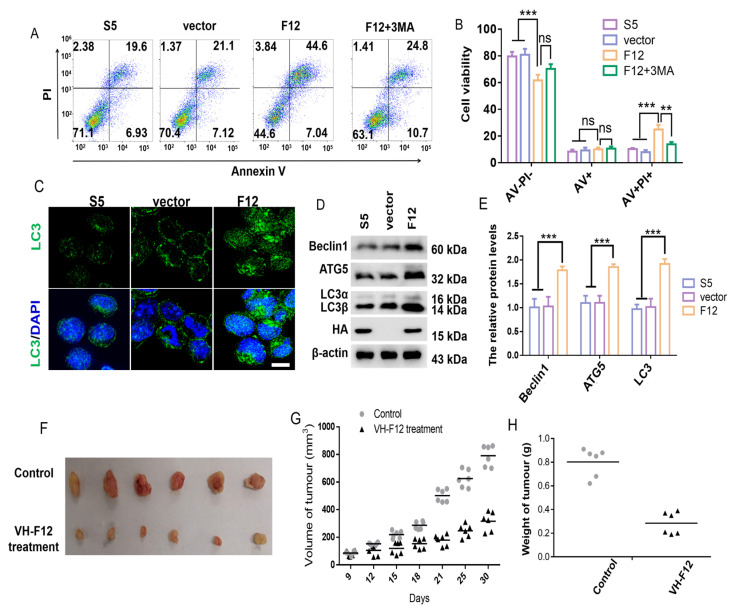
Effect of VH-F12 intrabody on antitumoral efficacy. (**A**) Flow cytometry analysis of the portion of Annexin V + (AV+) and propidium iodide (PI +) cells in MKN-45 cells. Cells transfected with pcDNA3.1(-)-VH-F12 were treated with 10 mM autophagy inhibitor (3-Methyladenine, 3-MA) for 24 h. (**B**) Results are the means ± S.E.M. of three independent experiments, each performed in duplicate. ns, no significance, ** *p* < 0.01, *** *p* < 0.001; by student’s *t*-test. (**C**) Immunofluorescence observation of LC3+ structures (green) in GC cells following transfection with pcDNA3.1(-)-VH-F12, pcDNA3.1(-)-VH-irrelevant (S5) or vector (each point representing the number of LC3 + vesicles in a cell) (scale bar, 10 µm). (**D**,**E**) Western blot showing expression levels of Beclin 1, ATG5, and LC3α, β. Bar graphs were obtained by normalizing to actin. The results are the means ± S.E.M. of three independent experiments, each performed in duplicate. *** *p* < 0.001; by student’s *t*-test. (**F**) MKN-45 xenografts grown in nu/nu mice received multipoint i.t. injections of 1 × 10^7^ TU Lenti-VH-F12 suspended in a total volume of 100 μL on day 9 and day 12 (*n* = 6). Negative control mice (*n* = 6) received a dose equivalent to 1 × 10^7^ TU negative control lentivirus given in identical volume. (**G**) The effect of VH-F12 intrabody on tumor volume at different time points. (**H**) The effect of VH-F12 intrabody on tumor weight. MKN-45 xenograft mice were treated for 4 weeks and tumor tissues were removed and weighed. Data are shown as the mean ± S.E.M. of 6 mice by student’s *t*-test (** *p* < 0.01, *** *p* < 0.001).

**Figure 5 ijms-24-14787-f005:**
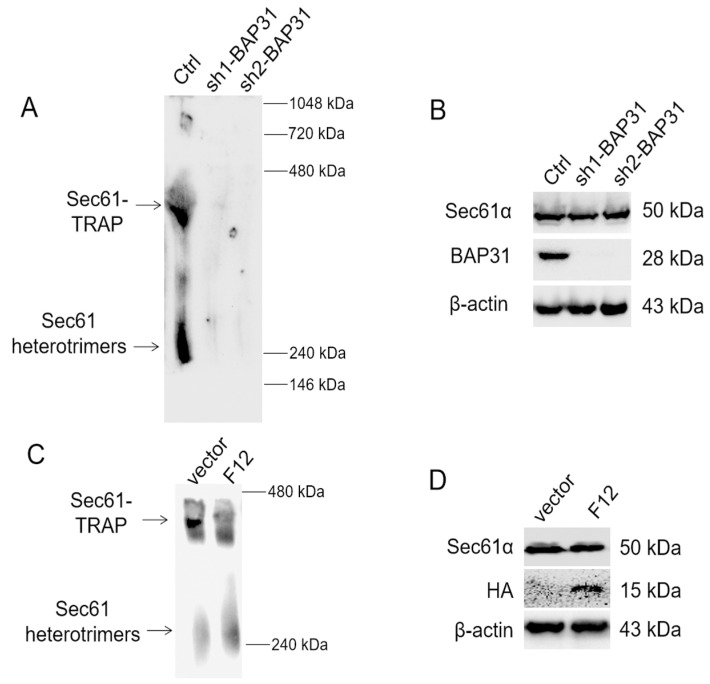
The formation of the Sec61 and Sec61-TRAP complexes in BAP31 knockout or VH-F12 transfected cells. (**A**) BN-PAGE of protein from WT MKN-45 cells (lane 1) or BAP31 knockout cells (lane 2 and 3) were analyzed by protein immunoblotting. Anti-sera to Sec61α recognized the Sec61-TRAP complex and Sec61 heterotrimers. (**B**) The protein level of Sec61α and BAP31 by SDS-PAGE in WT MKN-45 cells (lane 1) or BAP31 knockout cells (lanes 2 and 3). (**C**) BN-PAGE of protein from WT MKN-45 cells (lane 1) or VH-F12 transfected cells (lane 2) were analyzed by protein immunoblotting. (**D**) The protein level of Sec61α and BAP31 by SDS-PAGE in WT MKN-45 cells (lane 1) or VH-F12 transfected cells (lane 2).

**Figure 6 ijms-24-14787-f006:**
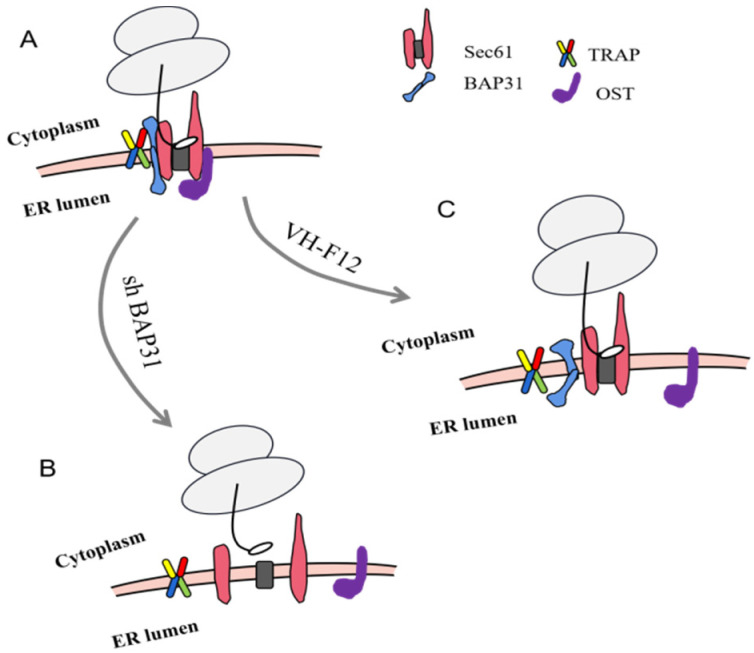
The simulation diagram of BAP31 participating in the process of EpCAM glycosylation. (**A**) The Sec61 translocon and BAP31 mediates EpCAM transport into the ER. (**B**) When BAP31 was knocked out, Sec61 complex could hardly be formed. (**C**) The incorrect three-dimensional conformation of BAP31 inhibits the interaction between Sec61 and TRAP complexes.

## Data Availability

The data that support the findings of this study are available from the corresponding author upon reasonable request.

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
