# Peer review of "An Intrabody against B-Cell Receptor-Associated Protein 31 (BAP31) Suppresses the Glycosylation of the Epithelial Cell-Adhesion Molecule (EpCAM) via Affecting the Formation of the Sec61-Translocon-Associated Protein (TRAP) Complex"

_ijms, 2023, doi:10.3390/ijms241914787_

Round 1

Reviewer 1 Report

Gastric cancer remains one of the cancer types with little increased life expectancy despite years of research into its therapy and novel molecular targets for gastric cancer is much needed in the field. In this study, the author focused on glycosylation of Epcam in gastric cancer and identified Bap31 as a potential targets for curbing gastric cancer via inhibition of N-glycosylation. The results can represent a significant step forward in gastric cancer therapy.

Some issues, however, must be addressed for this study. BAP31 activity depends on its cytoplasmic sequence 165-205 and the VH-F12 fragment was selected based on this region. In later tests of this intrabody, the intrabody was made to express in the ER lumen, which is separated from the cytoplasmic side by the membrane and cannot interact with the cytoplasmic domains of BAP31 in an intact cell. This is a serious flaw in the study design, as the intrabody is not expected to work if it has no access to its epitope sequence. The authors need to test a different construct, expressing the intrabody in the cytosol and compare the effect with the construct they have in this submission. Without a convincing result of intrabody acting from the cytoplasmic side, all the observed effect can at best be attributed to a mechanism irrelevant to the one proposed in the manuscript.

The manuscript is well written and may benefit from some minor edits to streamline the logic flow.

Author Response

Thanks very much for the question raised by the reviewer. These comments are very valuable and helpful for revising and improving our paper, as well as the important guiding significance to my researches. This is also a challenge we faced during the experimental process. Because we found in our previous experiment that the expression level of VH-F12 was very low to the extent that bands were almost undetectable by western blotting. At the same time, we found that F12 did not have any positive function.

We speculate that F12 may be rapidly degraded in the cytoplasm and therefore have not enough time to effectively bind to BAP31. So we attempted to add signal peptides with the aim of guiding F12 into the ER lumen. Due to the fact that the translation, folding and maturation of BAP31 occurs within the ER lumen, guiding F12 entering the ER lumen could greatly increase its chances of binding to BAP31 (165-205). The experimental results confirm our hypothesis. The results showed that the addition of signal peptides enabled stable expression of F12 in cells and promoted F12 to effectively combine with BAP31 (165-205) ( Figure 3A ). Furthermore, our subsequent experimental results support our design for introduing signal peptides. The addition of signal peptides stabilized F12, promoted the binding of F12 to BAP31(165-205) , and thereby exerting its function of inhibiting gastric cancer.

The issues raised by the reviewer have sparked a rethinking of our mechanism. The new explanation is as follows: the combination of F12 with BAP31 in the ER lumen may affect the correct folding of BAP31 and thereby inhibiting the interaction between Sec61 and TRAP complexes. We have made corresponding modifications to the discussion section (line348-350) and schematic diagram (Figure 6).

We appreciate for your warm work earnestly and hope that the correction will meet with approval. Thanks very much for your time and consideration and we greatly look forward to hearing from you.

Sincerely yours,

Bing Wang

Reviewer 2 Report

The authors present a very careful and well designed study on the role of BAP31 for glycosylation of EpCAM. The molecular interaction was demonstrated by coimmunoprecipitation and immunofluorescence. Flow cytometry and Western analysis were used as tools for determining protein expression levels. The role of BAP31 for glycosylation of EpCAM were demonstrated by band shift assays using glycosidase and specific site mutations of EpCAM. A single domain human antibody was used for the design of an intrabody interfering with BAP31-EpCAM interaction suppressing glycosylation of EpCAM and thus inducing cell death. Cell death was found not to be apoptotic, but rather associated with the activation of autophagy. The intrabody inhibiting glycosylation of EpCAM was demonstrated to interfere with tumor growth in vivo. In conclusion, this is a really impressive work demonstrating the molecular function of BAP31 for the ER complex Sec61-TRAP responsible for rotein translocation into the ER. The methods are well described, the data well presented. I am missing an important reference: PMID: 24898727, demonstrating the role of BAP31 for EpCAM function in stem cells. After including that, I think this study deserves publication.

Minor points. I would use te official gene name BCAP31 instead of BAP31, the short name of the protein. The authors might reconsider their title focussing on the anti-cancer effects of the anti-BAP31 intrabody. While this is ok, I think the elaboration of the molecular functions of BAP31 for the Sec61-TRAP complex and glycosylation of EpCAM are at least as relevant and maybe of more general interest.

Author Response

     Thanks very much for the reviewer's valuble comments.

     Question 1: I would use te official gene name BCAP31 instead of BAP31, the short name of the protein.. 

      Response: the gene name of BAP31 has been corrected to BCAP31.

      Question 2. The authors might reconsider their title focussing on the anti-cancer effects of the anti-BAP31 intrabody. While this is ok, I think the elaboration of the molecular functions of BAP31 for the Sec61-TRAP complex and glycosylation of EpCAM are at least as relevant and maybe of more general interest.

      Response:The title has been corrected as  "An intrabody against BCAP31 suppresses the glycosylation of EpCAM via affecting the formation of Sec61-TRAP complex"

Round 2

Reviewer 1 Report

The authors did not address my concerns and only made minor changes to the text.

Author Response

Dear reviewer,

We are very sorry that our response was not very satisfactory to you.

We have tested the construct of VH-F12 to make it express in the cytosol previously. But the expression level of VH-F12 was very low in this situation to the extent that the bands were almost undetectable by western blotting. We also investigated the inhibitory effect and antitumor activity of VH-F12 expressing in the cytosol and found that it did not have any positive function. This is a major challenge we face during the early experimental process. We speculated that F12 may be rapidly degraded in the cytoplasm, so there is not enough time to effectively bind to BAP31, resulting in very low expression levels and inability to function in the cell.

To solve this difficult problem, we attempted to add signal peptides with the aim of guiding F12 into the ER lumen. Due to the fact that the translation, folding and maturation of BAP31 occurs within the ER lumen, guiding F12 entering the ER lumen could greatly increase its chances of binding to BAP31 (165-205). The experimental results confirmed our hypothesis. The results showed that the addition of signal peptides enabled stable expression of F12 in cells and promoted F12 to effectively combine with BAP31 (165-205) ( Figure 3A ). Furthermore, our subsequent experimental results support our design for introduing signal peptides to guide F12 into the ER lumen. The addition of signal peptides stabilized F12, promoted the binding of F12 to BAP31(165-205) , and thereby exerting its function of inhibiting gastric cancer.

We have corrected the mechanism by which VH-F12 may function. The new explanation is as follows: the combination of F12 with BAP31 in the ER lumen may affect the correct folding of BAP31 and thereby inhibiting the interaction between Sec61 and TRAP complexes. We have made corresponding modifications to the discussion section (line348-350) and schematic diagram (Figure 6).

We sincerely thanks for your kind enthusiasm and helpful comments regarding our paper.

Sincerely yours,

Bing Wang
